# Evaluating the Molecular Properties and Function of ANKHD1, and Its Role in Cancer

**DOI:** 10.3390/ijms241612834

**Published:** 2023-08-16

**Authors:** Jordan L. Mullenger, Martin P. Zeidler, Maria Fragiadaki

**Affiliations:** 1Department of Infection, Immunity, and Cardiovascular Disease, The University of Sheffield, Sheffield S10 2RX, UK; jlmullenger1@sheffield.ac.uk; 2Department of Translational Medicine and Therapeutics, Queen Mary University London, London E1 4NS, UK; 3School of Biosciences, The University of Sheffield, Sheffield S10 2TN, UK; m.zeidler@sheffield.ac.uk

**Keywords:** ANKHD1, ankyrin repeats, KH domain, proliferation, cancer, RNA binding, post-translational modifications

## Abstract

Ankyrin repeat and single KH domain-containing protein 1 (ANKHD1) is a large, scaffolding protein composed of two stretches of ankyrin repeat domains that mediate protein–protein interactions and a KH domain that mediates RNA or single-stranded DNA binding. ANKHD1 interacts with proteins in several crucial signalling pathways, including receptor tyrosine kinase, JAK/STAT, mechanosensitive Hippo (YAP/TAZ), and p21. Studies into the role of ANKHD1 in cancer cell lines demonstrate a crucial role in driving uncontrolled cellular proliferation and growth, enhanced tumorigenicity, cell cycle progression through the S phase, and increased epithelial-to-mesenchymal transition. Furthermore, at a clinical level, the increased expression of ANKHD1 has been associated with greater tumour infiltration, increased metastasis, and larger tumours. Elevated ANKHD1 resulted in poorer prognosis, more aggressive growth, and a decrease in patient survival in numerous cancer types. This review aims to gather the current knowledge about ANKHD1 and explore its molecular properties and functions, focusing on the protein’s role in cancer at both a cellular and clinical level.

## 1. Structure and Function of ANKHD1

Ankyrin repeat and single KH domain-containing protein 1 (ANKHD1) is a large protein encoded by the 147 kbp *ANKHD1* gene located on human chromosome 5q31.3 (139,781,399–139,929,163 bp) and is ubiquitously expressed in all major organs [1]. ANKHD1 shares high sequence homology with its *Drosophila* ortholog- multiple ankyrin repeats single KH domain (Mask)- Figure 1. Overall, the proteins confer 74% similarity [2]. Mask is expressed in the developing eyes of *Drosophila melanogaster*, where it decreases apoptosis whilst increasing cellular proliferation and differentiation [3]. This review will examine recent data on how ANKHD1 may drive cellular proliferation and differentiation in the same way as its orthologue Mask, specifically focusing on its role in cancer.

## 2. The Protein Domains of ANKHD1

ANKHD1 contains 25 ankyrin motifs grouped into two domains, and a single KH (K homology) domain [3]. Compared to Mask, the 1st set of ankyrin domains has a 79% sequence similarity, the 2nd set of ankyrin domains has a 63% sequence similarity, and the KH domain has a 54% sequence similarity [2].

The ankyrin repeat motif is highly conserved and is made up of 33 amino acids, which were first identified in the *Drosophila* signalling proteins Notch and the yeast cell cycle regulators Swi6 and Cdc10 [6]. These motifs explicitly mediate protein–protein interactions [7], which led to the original suggestion that ANKHD1 acts as a scaffolding protein [8] that interacts within signalling pathways to mediate diverse functions. Over time, additional studies have reinforced the scaffolding role of ANKHD1 by identifying protein interactions between ANKHD1 and components of crucial signalling cascades, such as the JAK/STAT [9], Hippo [10,11], and the PINK/PARKIN pathways [12]. Yeast two-hybrid studies have also identified that the ankyrin domains of ANKHD1 (specifically from amino acid 1130–1243) bind to the apoptotic factor SIVA1 to inhibit its function [13].

ANKHD1 also contains a single KH domain. The KH domain is a short amino acid sequence that mediates binding to single-stranded ribonucleic acids (RNAs) such as microRNAs (miRNAs), mRNA (messenger RNAs) or single-stranded DNA (ssDNA) [14,15], classifying ANKHD1 as an RNA binding protein. ANKHD1 has been shown to interact with three miRNAs- miR-29a-3p, miR-205, and miR-196a [16] and the long non-coding RNA LINC00346 [17]. Given the presence of both ankyrin and KH domains, it is clear that ANKHD1 can interact with both proteins and RNAs, giving it the potential to mediate a significant number of potentially meaningful interactions.

As no crystal structure has been produced for ANKHD1, a predicted model of the full-length protein has been generated using AlphaFold [18,19] (Figure 2). This predicted ANKHD1 structure has high levels of confidence in the organisation of the ankyrin repeats and KH domain, as the configuration of these domains is highly conserved across proteins. However, the disordered regions are inherently poorly characterised, and the AlphaFold shows low levels of confidence. The ankyrin domains of ANKHD1 are coiled into an alpha helix, creating extensive opportunities for bonding and strong interactions between ANKHD1 and its interacting proteins. The KH domain is located less centrally to the structure, potentially allowing for easier access by associating with partner RNAs.

## 3. Isoforms of ANKHD1

In total, 34 exons make up the *ANKHD1* coding region [20], which can undergo alternative splicing to produce multiple transcript variants [8] with a diverse range of functions. From these multiple transcript variants, there are currently 22 identified isoforms of ANKHD1 [21], but likely many more have yet to be discovered. Isoform 1 (referred to as ANKHD1-203, ENST00000360839.7 in Ensembl [22] and Q8IWZ3-1 in Uniprot [4]) is the full-length ANKHD1 protein, made up of 2542 amino acids and with a molecular weight of 270 kDa; it is the most abundant and best characterised of the isoforms [13].

Isoforms 2, 3, and 4 (Figure 3A) are 616, 627, and 581 amino acids in length, respectively, and are significantly smaller than isoform 1 (Figure 3B) due to premature termination after the 11th ankyrin motif in the first block of ankyrin repeats (Appendix A). Additionally, isoform 2 contains a deletion of 11 amino acids at position 152, which truncates the protein further. Of these three isoforms, the best characterised is isoform 3, also referred to as HIV-1 Vpr binding ankyrin repeat protein (VBARP), which is located in the cytoplasm and has a known anti-apoptotic role in cell survival through the regulation of caspases [23]. However, compared to isoform 1, these transcript variants are much more poorly characterised, and little is known about the isoform-specific functions and subcellular localisations of isoforms 2 and 4 and whether or not they form functional proteins within cells. Isoform 5 is the outcome of the readthrough of the *ANKHD1* gene and part of the downstream locus *EIF4EBP3*, an event which produces a fusion transcript comprised of the ANKHD1 protein sequence and the C-terminus of EIF4EBP3, called ANKHD1-EIF4EBP3 (Figure 3A,B). It is notable that the readthrough is evolutionarily conserved, hence likely to have evolutionarily important functions, with its orthologous fusion protein known as Mask-BP3 in *Drosophila* [8]. The neighbouring gene, *EIF4EBP3,* encodes a member of the elongation initiation factor, the EIF4EBP family, which binds to the eukaryotic translation initiation factor 4E [24]. ANKHD1-EIF3EBP3 is expressed across a wide range of mammalian tissues [4] as detected via RNA sequencing, yet the function of this protein fusion and the conditions that promote its expression, are currently unknown. More generally, it is hypothesised that gene fusions, derived via transcriptional readthrough, drive protein evolution [25], allowing for diverse functions under conditions of stress.

Of the 22 ANKHD1 isoforms, 7 retain at least one intron, predicted to lead to protein degradation via the nonsense-mediated decay pathway, producing little or no detectable protein [27]. The specific intron retained varies between transcripts, and the resulting products have no known functions. The remaining ten transcripts listed by Ensembl [22] and Uniprot [4] are computationally mapped potential isoform sequences and have not been experimentally validated.

The multiple isoform lengths of ANKHD1 are observed as a ladder of bands appearing on a Western blot when probing with a specific anti-ANKHD1 antibody (Figure 4).

## 4. Post-Translational Modifications of ANKHD1

Some of the bands observed when ANKHD1 is visualised on a Western blot (Figure 4) can be explained by the protein’s alternatively spliced isoforms and readthrough expression; however, not all of them match the predicted molecular sizes. One potential explanation is post-translational modifications (PTMs), which can alter the size and the electrophoretic migration characteristics of proteins, resulting in the characteristic ‘laddering’ effect observed in the Western blots of ANKHD1.

We used PhosphoSitePlus^®^ [31], an open-source curated database that collates experimentally observed PTMs from low and high throughput studies, to predict PTMs in ANKHD1 and mapped them onto the protein sequence (Figure 5).

There are 48 predicted phosphorylation sites, and whilst the functional significance and dynamics of these sites are unclear, ANKHD1 is known to interact with multiple signalling pathways, which are themselves regulated via protein phosphorylation. As such, we speculate that ANKHD1 phosphorylation (and other modifications) could regulate these pathways in a signal-responsive manner. Three of the phosphorylation events predicted using PhosphoSitePlus^®^ affect tyrosine residues and allow for the potential phosphorylation of ANKHD1 via a tyrosine kinase. However, the majority of phosphorylation events identified in ANKHD1 affect serine and threonine residues, the kinases of which are activated in response to a range of biological signals, including DNA damage and prolonged cellular stress.

In addition to phosphorylation, seven acetylation sites, five methylation and four ubiquitination sites have also been identified [31] as potential PTMs of ANKHD1 (Figure 5), although their potential functions have not yet been investigated, and so their role is unknown.

Taken together, PTMs alter the weight of proteins and target ANKHD1, which could explain some of the laddering observed on the Western blot for ANKHD1 (Figure 4), not accounted for by alternative isoforms, but further work is needed to identify the functional significance of these interesting modifications.

## 5. Subcellular Localisation of ANKHD1

Since the characterisation of ANKHD1 began in 2005, its subcellular localisation has been unclear. Structurally, ANKHD1 contains no signal peptides or transmembrane domains. However, recent evidence does demonstrate that ANKHD1 is associated with a part of the early endosome and has a role in the fission of the membrane bilayer [32].

Early histochemical studies into the localisation of ANKHD1 were performed in a range of cell types, including HeLa cells, multiple myeloma cell lines and leukaemia cell lines. Using laser confocal microscopy, the protein was demonstrated to be located explicitly in the cytoplasm and for some time, the protein was believed to only function within that subcellular compartment [23,33,34]. Extensive roles for ANKHD1 have been identified in the cytoplasm, including the role of ANKHD1 in SHP2 signalling [33], the JAK/STAT signalling pathway at the plasma membrane [9,35], and a role for isoform 3 of ANKHD1 in the regulation of caspases for cellular survival and apoptosis [23]. Additionally, many of the roles of ANKHD1 in cancer were identified in the cytoplasm including multiple myeloma [34] and leukaemia [33] cell lines.

More recently, in 2015, Dhyani et al. performed a chromatin immunoprecipitation (ChIP) analysis, showing that ANKHD1 was also localised within the nucleus, where it associates with DNA to repress the p21 promoter by directly binding to it in a multiple myeloma cell line. This study was the first to suggest that ANKHD1 may translocate into the nucleus, a hypothesis supported by the use of a nuclear export inhibitor (Leptomycin B) and the observed accumulation of ANKHD1 in the nucleus using confocal microscopy [36]. Additionally, ANKHD1 was demonstrated to interact with the promoter of the HSPB1 gene in the nucleus [37]. It is now widely accepted that ANKHD1 is most commonly localised to the cytoplasm of cells but can, at times, also be observed within the nucleus.

It has been suggested that ANKHD1 contains its own non-canonical nuclear export sequence, which allows for translocation between the nucleus and cytoplasm [32], although this nuclear export sequence has not been identified using current computational prediction models, nor was a specific DNA sequence identified as the nuclear export tag by the original authors of the hypothesis [36]. Whether ANKHD1 requires its own nuclear signal to enter the nucleus is unknown, although given its ability to form protein–protein complexes, it is likely that it translocates into the nucleus together with a protein partner, hence not needing its own import signal.

The potential functional significance of the nuclear localisation of ANKHD1 remains largely unknown, and there is insufficient evidence as to whether the subcellular localisation is tissue/cell type dependent, although it has been proposed [38].

## 6. Role of ANKHD1 in Signalling Pathways

### 6.1. SHP2 Signalling

Mask, the *Drosophila* ortholog of ANKHD1, was initially identified in a genetic screen for genes that would enhance the phenotype of a dominant negative form of Corkscrew (CSW) (human SHP2, a tyrosine phosphatase). Loss of Mask from this pathway resulted in a phenotype similar to the loss of components of the epidermal growth factor receptor (EGFR) signalling pathways, meaning Mask may be genetically interacting with components of the pathway; however, experiments have never been performed to validate which components. Mask was required for the differentiation of developing photoreceptors, especially the R1, R6, and R7, and driving photoreceptor survival and differentiation in the developing eye. Loss of Mask function resulted in smaller eye imaginal discs due to a reduction in imaginal disc cell differentiation and proliferation and an induction of apoptosis [3].

In humans, ANKHD1 and SHP2 have been demonstrated to physically interact via coimmunoprecipitation assays in leukaemia cell lines K562 and LNCap [33], and it was suggested that ANKHD1 may be functioning as an adaptor protein. However, ANKHD1 did not coimmunoprecipitate with SHP2 in other leukaemia cell lines (KG1, HL60, Daudi and Jurkat) despite them presenting with high levels of SHP2 protein expression, potentially demonstrating this interaction may be cell-line specific. The complete functional outcome of the potential Interaction of ANKHD1 with SHP2 remains elusive.

### 6.2. JAK/STAT Signalling Pathway

The Janus kinase (JAK)-signal transducer and activator of transcription (STAT) pathway is a complex signalling cascade, transducing signals from over 50 ligands, including cytokines and growth factors [39,40]. These ligands activate downstream genes that regulate immunity, haematopoiesis, and inflammation [41] in both embryonic development and adult pathophysiology. For example, SARS-CoV-2 infection results in inflammation, via activation of the JAK/STAT pathway, which is best known as the ‘cytokine storm’. JAK is a tyrosine kinase with both an active catalytic domain and an adjacent regulatory kinase-like domain, constitutively bound to the cytokine receptor. Upon ligand binding, the cytokine receptor dimerises or undergoes a conformational change that activates their associated JAKs, initiating autophosphorylation and the trans-phosphorylation of tyrosine residues on the receptor and downstream STAT. The phosphorylated STATs dimerise and translocate into the nucleus to interact with the promoter regions of genes and activate transcription (Figure 6) [9,42,43]. Both loss- and gain-of-function mutations in pathway components can result in a variety of diseases, including immunodeficiency and inflammatory conditions, including Crohn’s disease and blood cancers [44,45].

An initial genome-wide RNA interference screen performed to identify new genetic modulators of the JAK/STAT pathway in *Drosophila* identified Mask as a potential regulator of STAT transcriptional activity [9]. Follow-up research demonstrated that Mask affects the formation of cytokine receptor signalling by regulating receptor levels [35], with this increase in receptor activity resulting in an increase in JAK/STAT transcriptional activation. This was further supported by data from human studies demonstrating that the silencing of ANKHD1 resulted in a 50% reduction in STAT3 tyrosine phosphorylation [35] (Figure 6). Overall, it was demonstrated that ANKHD1 modulates JAK/STAT signalling in both humans and *Drosophila*, although whether ANKHD1 requires additional protein or RNA partners to exert its effects on this pathway is currently unknown.

### 6.3. Hippo Signalling Pathway

YAP (yes-associated protein), and its paralogue TAZ (transcriptional coactivator with PDZ-binding motif, also referred to as WWTR1), is a transcriptional coactivator that controls tissue growth in humans via the Hippo tumour suppressor pathway. When the Hippo signalling pathway is activated, the SAV1 (Salvador 1) and MST (mammalian sterile 20-related kinase) complex become phosphorylated, along with the MOB1 (MOB kinase activator) and LATS (large tumour suppressor kinase) complex. These upstream phosphorylation events drive the phosphorylation of YAP/TAZ, resulting in either its ubiquitination and degradation or its binding to the 14-3-3 protein and subsequent cytoplasmic retention. In both cases, YAP/TAZ is prevented from entering the nucleus, and hence, phosphorylation leads to pathway inactivation. Conversely, when the Hippo signalling pathway is active, upstream proteins are not phosphorylated, allowing YAP/TAZ to enter the nucleus and form a complex with one of the TEAD transcription factors (TEAD1-4), driving the transcription of genes with a role in proliferation, migration, and cellular survival (Figure 7) [46,47,48,49].

The second ankyrin domain and KH domain of ANKHD1 [10] were demonstrated to immunoprecipitate with YAP and TEAD in humans [11], with the resulting complex localised in both the nucleus [10,11] and cytoplasm [51]. ANKHD1 was demonstrated to be a positive regulator of YAP1 at both the protein level, by increasing the stability of YAP complexing with TEAD [52], and at the mRNA level regulating gene expression [53]; the presence of ANKHD1 results in an upregulation of the YAP mRNA, whereas the knockdown of ANKHD1 results in a downregulation of YAP mRNA [53]. Furthermore, ANKHD1 negatively regulates phosphorylated YAP1, limiting the amount of cytoplasmic retention and degradation of the protein. The YAP1/ANKHD1 complex has been demonstrated to drive epithelial-to-mesenchymal transitions (EMT) via the increased expression of vimentin and phosphorylated AKT and a reduction in the expression of E-cadherin [54].

The knockdown of ANKHD1 results in the decreased expression of YAP1 target genes, suggesting that not only does the complex drive cellular proliferation through gene transcription but that ANKHD1 is also required for the full activity of the YAP1/TEAD complex [10]. Moreover, increased complex formation of the ANKHD1/YAP complex has been identified in numerous cancer types due to downregulation of the Hippo signalling pathway, including non-small cell lung cancer [53], colorectal cancer [54], prostate cancer [51], and breast cancer [10].

### 6.4. p21 Signalling Pathway

p21 is a cyclin-dependent kinase inhibitor, predominantly associated with the inhibition of cyclin-dependent kinase 2 (CDK2) [55]; it promotes cell cycle arrest in response to DNA damage [56]. Chromatin coimmunoprecipitation (ChIP) revealed that the ANKHD1 protein physically interacts with the p21 protein [36]. Moreover, ChIP has also suggested that ANKHD1 may physically interact with the p21 promoter region in multiple myeloma cell lines [36]—suggesting two potential mechanisms of ANKHD1 and p21 interaction. Given the lack of DNA-binding motifs within ANKHD1, it is unlikely that the protein binds to the p21 promoter region directly and may associate via its protein binding partners. However, the knockdown of ANKHD1 with a short hairpin RNA (shRNA) increases p21 levels and results in cells arresting in the S phase [34]. Conversely, ANKHD1 overexpression decreased the levels of p21 bound to proliferating cell nuclear antigen (PCNA), causing a decrease in DNA repair and enhanced progression through the S phase—ultimately leading to uncontrolled cellular proliferation (Figure 8) [57].

## 7. Role of ANKHD1 in Cancer

Cancer is characterised by crucial hallmarks, including uncontrolled proliferation, cellular invasion, and metastasis [58,59]. As mentioned above, ANKHD1 was identified in a screen for proteins which interact with SH2-containing protein–tyrosine phosphatase (SHP2), a tyrosine phosphatase overexpressed in primary leukaemia cells and leukaemic cell lines [60]—an interaction that demonstrated the role of ANKHD1 in leukaemia. In 2006, Traina et al. published their study demonstrating that ANKHD1 has a role in regulating cell cycle proliferation in leukaemic cell lines and hypothesized that ANKHD1 may act to drive the proliferation of cancer cells [33]—a suggestion supported by the subsequent demonstration that increased ANKHD1 levels are sufficient to drive cellular proliferation whilst having no effect on apoptosis [13]. The previous roles of *Drosophila* Mask in the control of MAPK, JAK/STAT and Hippo are also supportive of its role in the control of cancer cell proliferation.

More recent papers provide further evidence of the role of ANKHD1 in solid cancers. Fragiadaki and Zeidler (2018) showed that ANKHD1 mRNA levels are upregulated in renal cell clear cell carcinomas, where it drives cellular division by engaging with several microRNAs, including miR-29a-3p [16]. This is the first report to demonstrate that ANKHD1 can modify cell behaviour via its RNA binding actions. More recently, ANKHD1 was also shown to interact with the long non-coding RNA (lncRNA) LINC00346 to regulate glioma angiogenesis [17] and with the lncRNA MALAT1 to promote resistance to radiotherapy in colorectal cancer [54]. While the entire repertoire of RNA interactors of ANKHD1 is currently unknown, this knowledge is likely to provide essential insights into the processes that take place during carcinogenesis.

Increased levels of ANKHD1 are also reported in prostate cancer cells [37], resulting in delayed cell cycle progression through the S phase and enhanced tumorigenicity in xenografts [51]. Furthermore, in multiple myeloma cells, ANKHD1 has been associated with promoting cell growth and division [34,36,57]. Colorectal cancer cells show increased levels of ANKHD1 and are associated with an increase in epithelial-to-mesenchymal transition, allowing increased cell mobility, possibly linking ANKHD1 to increased migration and the invasion of cancer cells [54]. Furthermore, increases in migration and invasiveness of cancer cells demonstrating high expression of ANKHD1 has also been described in hepatocellular carcinoma [61], breast cancer [38], non-small-cell lung cancer [53] and multiple myeloma [57] cells. Taken together, it is clear that increased expression of ANKHD1 is sufficient to sustain proliferation and growth [13,16,33,34,36,51,53,54,57,62] and to contribute to cellular invasion and metastasis [53,54,61,62] in multiple cancers.

Given that ANKHD1 overexpression is able to drive enhanced proliferation and cellular invasion of cancerous cells, it would be anticipated that overexpression of ANKHD1 may also impact cancer pathologies at a clinical level. This hypothesis was initially examined in breast cancer patients, where increased expression of ANKHD1 statistically correlates with a reduction in relapse-free survival (the length of time after the initial cancer treatment that the patient lives without any symptoms of that cancer [21]) [10,38], a change which varies depending on breast cancer subtype. Since this research, the clinical features correlating with increased levels of ANKHD1 have also been examined in colorectal cancer, non-small-cell lung carcinoma, and hepatocellular cancer. In all three studies, increased expression of ANKHD1 is associated with greater tumour growth, metastasis, larger tumours, and more nodules (abnormal tissue growths greater than 1 cm in diameter), resulting in poorer prognosis, more aggressive growth, and a decrease in patient prognosis [53,54,61]. This information is summarised in Table 1.

Analysis of mRNA expression levels of ANKHD1 in a range of both normal and paired cancerous tissue types reveals that ANKHD1 mRNA levels are increased in 13 of the 31 cancers examined (Figure 9A), with the largest increases in expression seen in oesophageal carcinoma, acute myeloid leukaemia, and stomach adenocarcinoma [63]. Of these 13 cancers, ANKHD1 is upregulated in specific subsets of glioma, renal cell carcinoma and leukaemia, which validates previous evidence that ANKHD1 is overexpressed in those cancer types [13,16,17,33]. However, most of the cancers do not show a significant change in ANKHD1 mRNA or mutation status in the cancer samples compared to the healthy tissue samples. In addition, the overall survival of patients with high ANKHD1 mRNA expression levels is poorer than those with low mRNA expression ANKHD1 levels between 200 and 300 months after diagnosis in a range of cancer types (Figure 9B).

Whilst the function of ANKHD1 initially appeared to be independent of cancer types, more recent studies have suggested that different transcript variants of ANKHD1 may confer different roles at both cellular and clinical levels. The transcript read-through variant ANKHD1-BP3 was shown to be highly expressed in uterine corpus endometrial carcinoma, with the knockdown of this transcript inhibiting cellular metastasis, invasion, apoptosis, and necrosis [37]. The role of ANKHD1-BP3 as an inhibitor of metastasis and invasion in uterine corpus endometrial carcinoma goes against the evidence of the role of ANKHD1, which is generally pro-proliferation, in other cancer types (such as breast cancer, renal cell carcinoma, leukaemia, etc.), for the first time suggesting the existence of ANKHD1 isoform-specific roles. Further work will need to be performed on the ANKHD1 isoforms to understand their individual functions.

**Table 1 ijms-24-12834-t001:** A summary of the cellular and clinical functions of ANKHD1 in cancer.

Type of Cancer	References	Cellular Function of ANKHD1	Clinical Features Associated with Enhanced ANKHD1
Breast Cancer	[10,38]	**Increases the viability, clonogenicity and migration of aggressive breast cancer cell lines, possibly due to the positive regulation of the YAP/Yki pathway.**	Increased levels of ANKHD1 were correlated with a reduction in relapse-free survival, but the extent differs based on the breast cancer subtype.
Colorectal Cancer	[54]	Increases cell proliferation, migration, and invasion. It also dysregulates the epithelial organisation of tissues by increasing epithelial-to-mesenchymal transition.	ANKHD1 expression was correlated with greater tumour infiltration. It was demonstrated to promote metastasis and growth in tumorous nodules.
Glioma	[17]	Promotes angiogenesis.	An increase in ANKHD1 is detected in glioma-associated endothelial cells, but there are no data to show how this may affect prognosis.
Hepatocellular Cancer	[61]	Enhances cellular migration and invasion.	Higher levels of ANKHD1 are correlated with larger tumours, more nodes, poorer differentiation of tumour boundaries, more advanced metastasis, shorter time between recurrence and overall reduced survival rates.
Leukaemia	[13,33]	Increases cellular and clonal proliferation and migration. It enhances tumorigenicity in xenografts.	Higher levels of ANKHD1 (protein and mRNA) are observed in leukaemia cell lines and (mRNA only) are observed in primary acute leukaemia samples.
Non-Small-Cell Lung Cancer	[53,62]	Stimulates proliferation, invasion, and colony formation. May have a role in promoting DDP-chemoresistance.	ANKHD1 expression was correlated with greater growth, metastasis, overall reduced survival rates and poorer prognosis. ANKHD1 is required for SMYD3-induced DDP chemoresistance.
Multiple Myeloma	[34,36,57]	Increases proliferation and cell-cycle progression. It enhances tumorigenicity in xenografts. It also promotes growth, division, and migration of cell colonies.	ANKHD1 is highly expressed in neoplastic plasma cells.
Pancreatic Ductal Adenocarcinoma	[64,65]	**Inferred control of cell growth by controlling the cell cycle inhibitor, p21.**	Premature termination of the ANKHD1 protein due to a nonsense mutation (E2410*) was correlated with Pancreatic Ductal Adenocarcinoma.
Prostate Cancer	[51]	Increases cell growth, cell cycle progression during S phase and enhances tumorigenicity in xenografts.	No data.
Renal Cell Carcinoma	[16]	Increases proliferation by upregulating proliferative genes and drives cellular division via mitosis.	ANKHD1 expression (protein and mRNA) is upregulated early in the kidneys of patients with renal cell carcinoma.
Uterine Corpus Endometrial Carcinoma	[37]	In murine models, transcript variant ANKHD1-BP3 was demonstrated to promote cell proliferation whilst inhibiting metastasis.	Transcript variant ANKHD1-BP3 was highly expressed in primary tissue samples. In clinical samples, it was demonstrated to promote proliferation but inhibit metastasis, invasion, apoptosis, and necrosis of cells.

**Figure 9 ijms-24-12834-f009:**
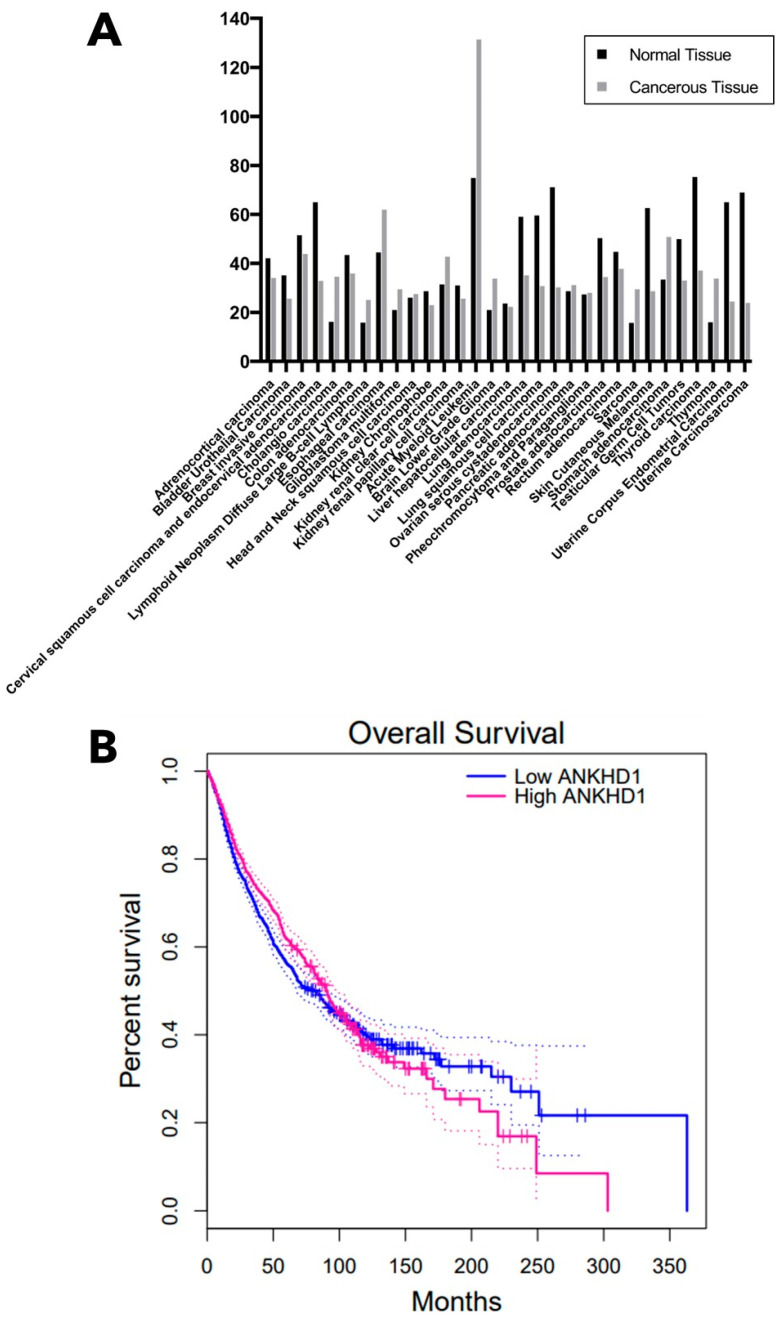
mRNA Expression of ANKHD1 in Healthy and Cancerous Tissue. (**A**) mRNA expression of ANKHD1 in multiple healthy and paired cancerous tissue types. The height of the bar represents media mRNA expression. (**B**) Overall patient survival in months after diagnosis with cancer in patients with high ANKHD1 mRNA levels (magenta) and low ANKHD1 mRNA levels (blue). High mRNA levels are defined as those in the upper quartile of all ANKHD1 mRNA expression levels, and low mRNA levels are defined as those in the upper quartile of all ANKHD1 mRNA expression levels. n = 2334. mRNA expression data and patient survival data were acquired from the TCGA and GTEx databases via GEPIA [63].

## 8. Summary and Concluding Remarks

ANKHD1 is a large scaffold protein containing ankyrin repeat domains which mediate protein–protein interactions and a KH domain that mediates RNA binding. As such, ANKHD1 is shown to be a critical scaffold able to coordinate and integrate essential interactions, including tissue growth. ANKHD1 is known to form a complex with proteins in a number of crucial signalling pathways, including receptor tyrosine kinase signalling [3,33], the JAK/STAT pathway [9], the Hippo signalling pathway [10,11], and p21 [34,36,57]. However, the full range of ANKHD1 functions across signalling pathways has yet to be determined. 

ANKHD1 has five known, stable isoforms, including the full-length protein, a readthrough isoform and three truncated isoforms; however, it is likely that there are additional unidentified isoforms of ANKHD1. The multitude of ANKHD1 isoforms provide an explanation for some of the laddering observed on Western Blots of ANKHD1 (Figure 4) but do not explain all the identifiable bands. This review suggests that the observed banding could also be the consequence of post-translational modifications to the proteins, which frequently change the charge and molecular weight of the modified proteins. Another explanation for the additional bands of ANKHD1 could be the result of degradation and/or proteolytic processing of the protein in the cell. However, there is no definitive evidence in the literature to support either hypothesis and more work will need to be done to determine the identity of all the additional bands, some of which may be the computationally predicted isoforms. Studies on the role of ANKHD1 in cancer cell lines demonstrate that the protein has a role at the cellular level in driving uncontrolled cellular proliferation and growth, causing enhanced tumorigenicity, cell cycle progression, and increased epithelial-to-mesenchymal transition in cancer cells with known oncogenic mutations. Furthermore, at a clinical level, the increased expression of ANKHD1 has been demonstrated to result in a reduction in the relapse-free survival of patients, greater tumour infiltration, increased metastasis, larger tumours, resulting in poorer prognosis, more aggressive growth, and a decrease in the survival rate of patients, regardless of the type of cancer [53,54,61]. However, the read through isoforms of ANKHD1 may have opposite roles. Further research into whether ANKHD1 could be a proliferative driver in other disease types is however yet to be investigated.

## Figures and Tables

**Figure 1 ijms-24-12834-f001:**
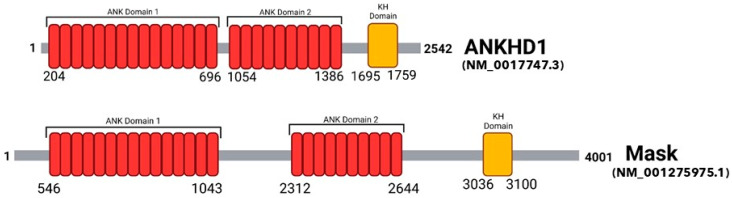
Comparison of the Domains in the Primary Structure of human ANKHD1 and Drosophila Mask. A schematic representation comparing the two ankyrin repeat domains (red) and the KH domain (yellow) located in the primary structure of ANKHD1 (*Homo sapiens*; isoform 1; NM_017747.3) and Mask (*Drosophila melanogaster*; NM_001275975.1). The amino acid number at which the protein initiates and terminates, and the initiation and termination of each domain, are displayed. Domain structure obtained from the UniProt database [4], and Prosite [5].

**Figure 2 ijms-24-12834-f002:**
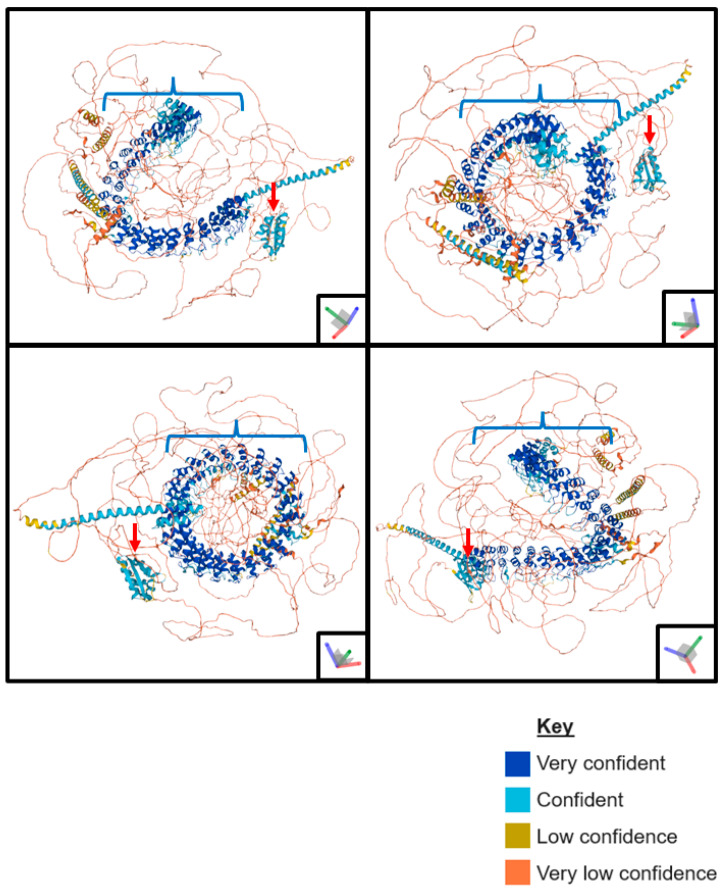
Predicted AlphaFold Tertiary Structure of ANKHD1. A schematic representation of the AlphaFold predicted tertiary structure of ANKHD1 showing its folding in 3D space, observed from 4 different angles. The blue brackets indicate the ankyrin repeat domains coiled into an alpha helix, and the red arrows indicate the KH domain. AlphaFold generates a per-residue confidence score (pLDDT) of 0 and 100, with 0 being no confidence and 100 being highly confident. In this diagram, very high is a pLDDT > 90, dark blue, confidence is a pLDDT > 70, light blue, low confidence is a pLDDT > 50, yellow, and very low confidence is a pLDDT < 50, orange. If a series of residues has consecutively low confidence, they may be indicated as a line on the diagram—images generated using AlphaFold [18,19].

**Figure 3 ijms-24-12834-f003:**
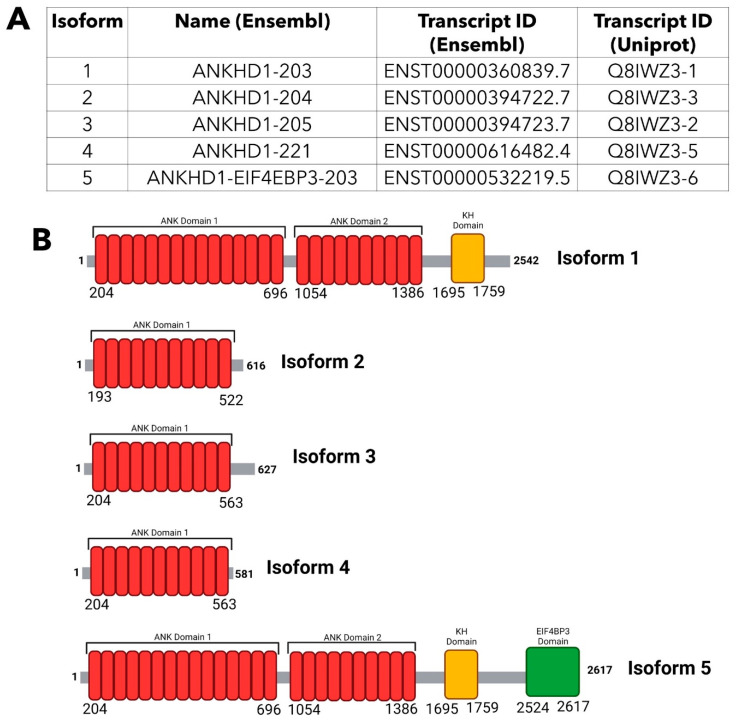
Comparison of the Isoforms of ANKHD1. (**A**) Details of the main five isoforms of ANKHD1, including their name and transcript ID by Ensembl [22] and the transcript ID by Uniprot [4]. (**B**) A schematic representation comparing the main 5 isoforms of ANKHD1. The two ankyrin repeat domains are indicated in red, the KH domain is indicated in yellow, and the EIF4BP3 domain is indicated in green. The amino acid number at which the protein initiates and terminates, and the initiation and termination of each domain, are displayed. The data for figure construction was obtained from Ensembl [22], UniProt [4] and NCBI databases [26].

**Figure 4 ijms-24-12834-f004:**
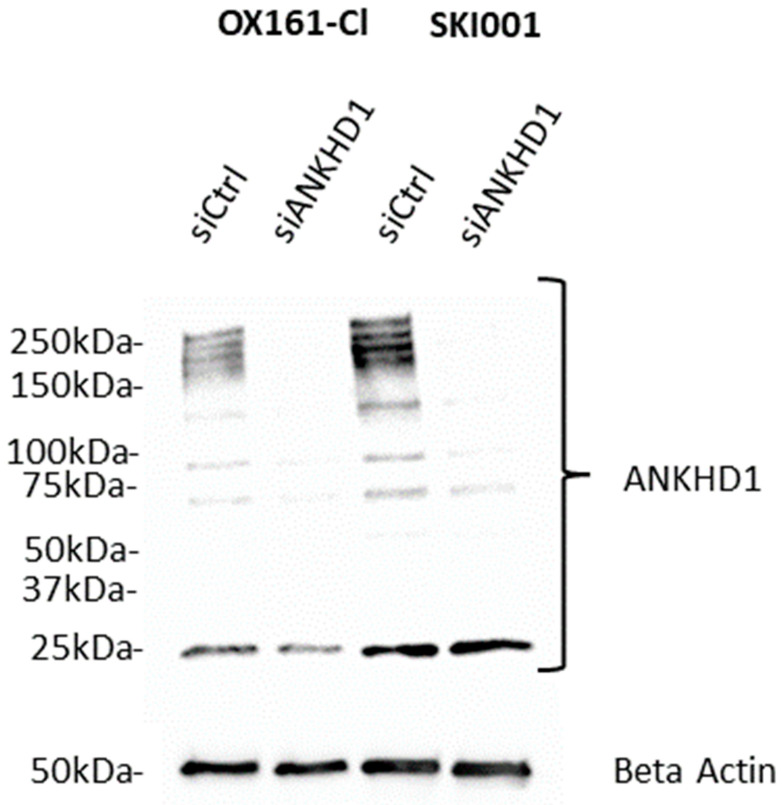
Western Blot to Detect Human ANKHD1. A representative Western Blot of ANKHD1 when whole cell lysate from either OX161-Cl or SKI001 cells (described in [28]) is used after siRNA knockdown with either control siRNA (siCtrl) or ANKHD1 specific siRNA (siANKHD1). The primary antibodies used were affinity purified ANKHD1 (#HPA008718) antibody at 1:1000 dilution and visualised with anti-rabbit (#P0448) antibody, and beta-actin (#ab8224) antibody used as a loading control at 1:1000 dilution, visualised with anti-mouse (#P0447) antibody and developed using ECL select. The predicted band size of ANKHD1 is 270kDa, with some laddering expected [29,30]. There is a non-specific band at 25 kDa. Western blot created by Dr Fiona Macleod.

**Figure 5 ijms-24-12834-f005:**
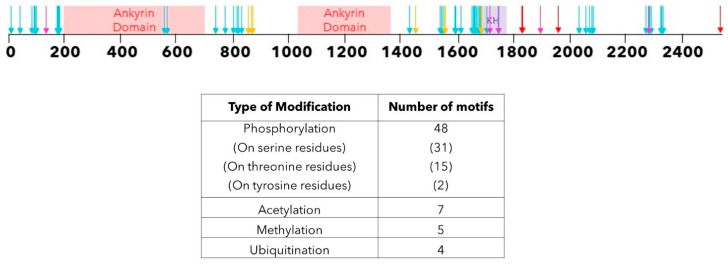
Post-translational Modifications of ANKHD1. A schematic representation of the locations of sequence motifs along the ANKHD1 protein, which may undergo PTM. The thin black line represents the amino acids of ANKHD1. The critical domains of ANKHD1 are indicated; red is for the ankyrin repeat domains, and purple is for the KH domain. The arrows indicate the presence of a specific PTM in that amino acid; cyan indicates phosphorylation sites, yellow indicates acetylation sites, magenta indicates methylation sites, and red indicates ubiquitination sites. Table summarises all the PTMs predicted PhosphoSitePlus^®^ [31].

**Figure 6 ijms-24-12834-f006:**
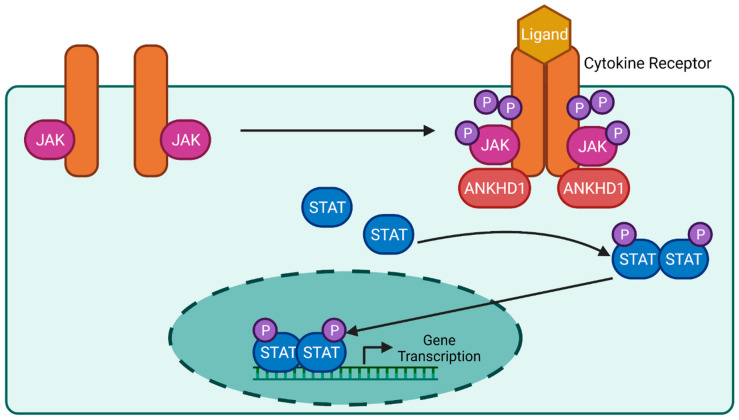
Predicted Role of ANKHD1 in the JAK/STAT Signalling Pathway. An overview of the critical components of the JAK/STAT signalling pathway in mammals. ANKHD1 is predicted to regulate the levels of cytokine receptors to allow for an increase in receptor levels and an increase in downstream JAK autophosphorylation. JAK autophosphorylation triggers the phosphorylation of the intracellular portion of the cytokine receptor and the phosphorylation of STAT dimers. This increase in phosphorylated STATs dimerisation results in increased translocation of the complex to the nucleus, where they interact with various promoter regions and causes an increase in gene transcription.

**Figure 7 ijms-24-12834-f007:**
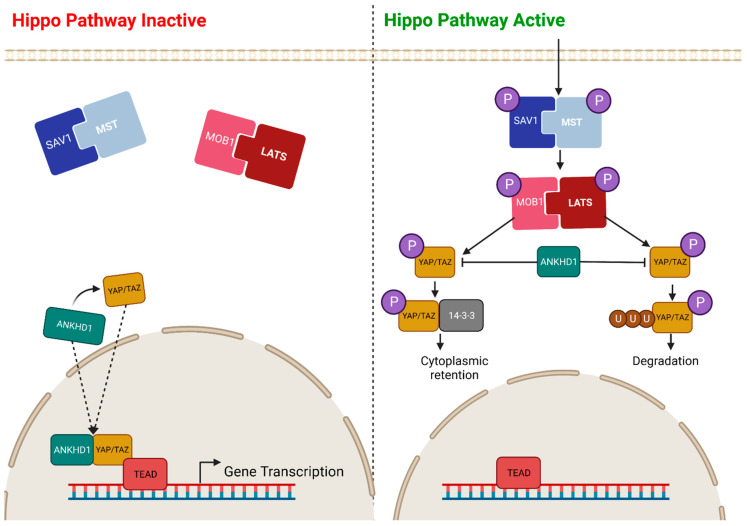
Predicted Role of ANKHD1 in the Hippo Signalling Pathway. A basic overview of the critical components of the Hippo signalling pathway. When the pathway is inactive (**left**), there is no upstream protein phosphorylation in the cytoplasm, allowing YAP/TAZ and ANKHD1 to form a complex and enter the nucleus. These proteins then form a complex with one of the TEAD transcription factors (TEAD1-4), driving gene transcription. Additionally, ANKHD1 positively regulates YAP mRNA. When the pathway is active (**right**), there is phosphorylation of both the SAV1/MST complex and the MOB1/LATS complex. This drives the phosphorylation of YAP, resulting in either ubiquitination and degradation or its binding to 14-3-3 and retention in the cytoplasm. Both cases prevent YAP and ANKHD1 from entering the nucleus, so gene transcription is inhibited [45,50]. However, ANKHD1 negatively regulates phosphorylated YAP, limiting the amount of cytoplasmic retention and degradation of the protein.

**Figure 8 ijms-24-12834-f008:**
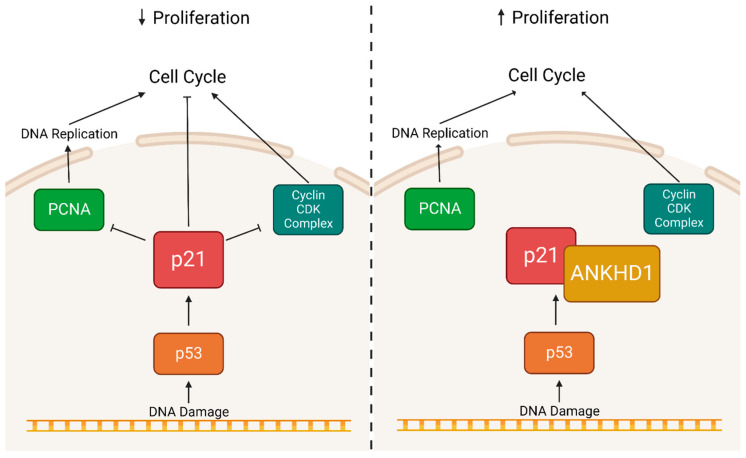
Predicted Role of ANKHD1 in the p21 Signalling Pathway. A basic overview of the critical components of the p21 signalling pathway in mammals is shown. DNA damage activates p53, which in turn activates p21, resulting in the inhibition of PCNA and a range of cyclin CDK complexes. This inhibition leads to the arrest of the cell cycle and the arrest of proliferation to prevent the propagation of DNA damage to additional cells. However, when ANKHD1 binds to p21, it prevents the inhibition of PCNA and cyclin CDK complexes, allowing for cellular proliferation even in the presence of DNA damage.

## Data Availability

Not applicable.

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
