# Peer review of "Evaluating the Molecular Properties and Function of ANKHD1, and Its Role in Cancer"

_ijms, 2023, doi:10.3390/ijms241612834_

Round 1
Reviewer 1 Report
Comments 1
The authors described that there are many papers on the full length of ANKHD1, but little is known about the function of its splicing variant. However, the splicing variant of ANKHD1 was first identified as an HIV-1 Vpr-binding protein in a yeast two-hybrid assay in 2005 (reference 1), and it has also been reported to be involved in the pathway of apoptosis.
Reference1
Molecular and functional characterization of a novel splice variant of ANKHD1 that lacks the KH domain and its role in cell survival and apoptosis Melissa C. Miles, Michelle L. Janket, Elizabeth D. A. Wheeler, Ansuman Chattopadhyay, Biswanath Majumder, Jeremy DeRicco, Elizabeth A. Schafer and Velpandi Ayyavoo. doi: 10.1111/j.1742-4658.2005.04821.x.
Comments 2
There is no description of the functions of individual domains. For example, Mask/ANKHD1 binds to Yki/YAP1 via ANK domain2 and KH domain from co-immunoprecipitation results (reference2). In a two-hybrid assay of yeast, it has been found to bind to SIVA1 at the 1130-1243aa portion of ANKHD1 (reference 3).
Reference2
Mask Is Required for the Activity of the Hippo Pathway Effector Yki/YAP. Leticia Sansores-Garcia, Mardelle Atkins, Ivan M. Moya, Maria Shahmoradgoli, Chunyao Tao, Gordon B. Mills, and Georg Halder. doi: 10.1016/j.cub.2012.12.033.
Reference3
ANKHD1 silencing inhibits Stathmin 1 activity, cell proliferation and migration of leukemia cells. João Agostinho Machado-Neto, Mariana Lazarini, Patricia Favaro, Paula de Melo Campos, Renata Scopim-Ribeiro, Gilberto Carlos Franchi Junior, Alexandre Eduardo Nowill,Paulo Roberto Moura Lima, Fernando Ferreira Costa, Serge Benichou,Sara Teresinha Olalla Saad, Fabiola Traina. doi: 10.1016/j.bbamcr.2014.12.012.
Comments 3
Figure 2 shows AlphaFold's prediction of the three-dimensional structure of ANKHD1, but there appeared to be no function of ANKHD1 from this structure. Some thoughts should be described from the structural prediction.
Comments 4
Since this review is about the role of ANKHD1 in cancer cells, the function of ANKHD1 in the nucleus is very well described in the text. However, the function of ANKHD1 in the cytoplasm is poorly written. Several cytoplasmic roles of ANKHD1 in relation to cancer cells have been reported.
Minor comments
1, Figure 1, Please describe the Accession number of Mask.
2, How homologous is MASK to ANKHD1?
3, Figure 3 shows the domain structure of isoform2, isoform3, and isoform4. However, these isoforms are shown as 15 ankyrin repeats like ANK domain 1, even though they obviously have shorter amino acid sequences than ANK domain of isoform 1?
4, Figure 4 shows a western blot of ANKHD1. The text says that isoform 5 appears at the highest, but it is not clear whether the expression of the isoform5 has actually been observed in OX161-CI and SKI001 cells, i.e., western blots alone will not be enough to show the presence of specific isoforms.
5, Figure 5 shows the phosphorylation site of ANKHD1, and most of the phosphorylation sites are located in the disorder region. Is there anything we know about the functional control of ANKHD1 by phosphorylation?
6, The legends in Figure 6 and Figure 7 do not describe ANKHD1. I think it would be better if the explanation of ANKHD1 was in legend.
Reviewer 2 Report
I have some suggestions:
1) Fig. 5C is unlikely necessary or at most should be shortened and only show the site with multiple HTPs
2) Line 192, homologue to ortholog to be consistent
3) What's the full name for Mask
4) Section 6.3 line 243 The TAZ here is NOT tafazzin. It is called: Transcriptional co-activator with PDZ-binding motif or WWTR1.
5) Line 315, Hippo not HIPPO.
6) In Section 7, may be better to include a Pan-cancer analysis for ANKHD1 mRNA expression with corresponding normal tissue using existing databases like TCGA.
